

# Low-level jets over the North Sea based on ERA5 and observations: together they do better

Peter C. Kalverla[1], James B. Duncan Jr.[2], Gert-Jan Steeneveld[1], and Albert A.M. Holtslag[1]

[1]Wageningen University, Meteorology and Air Quality Section, PO Box 47, 6700AA Wageningen, The Netherlands
[2]ECN part of TNO, Wind Energy, Westerduinweg 3, 1755LE Petten, The Netherlands

**Correspondence:** Peter Kalverla (peter.kalverla@wur.nl)

**Abstract.** Ten years of ERA5 reanalysis data are combined with met-mast and LiDAR observations from ten offshore platforms to investigate low-level jet characteristics over the Dutch North Sea. The objective of this study is to combine the best of two worlds: (1) ERA5 data with large spatiotemporal extent but inherent accuracy limitations due to a relatively coarse grid and an incomplete representation of physical processes, and (2) observations that provide more reliable estimates of the measured quantity, but are limited in both space and time. We demonstrate the effect of time and range limitations on the reconstructed wind climate, with special attention paid to the impact on low-level jets.

For both measurement and model data, the representation of wind speed is biased. The limited temporal extent of observations leads to a wind speed bias on the order of $\pm 1$ m s$^{-1}$. In part due to data-assimilation strategies that cause abrupt discontinuities in the diurnal cycle, ERA5 also exhibits a wind speed bias of approximately 0.5 m s$^{-1}$. Representation of low-level jets in ERA5 is poor in terms of a one-to-one correspondence, and the jets appear vertically displaced ('smeared out'). However, climatological characteristics such as the shape of the seasonal cycle and the affinity with certain circulation patterns are represented quite well, albeit with different magnitudes. We therefore experiment with various methods to adjust modelled low-level jet rate to the observations or, vice versa, to correct for the erratic nature of the short observation periods using long-term ERA5 information. While quantitative uncertainty is still quite large, the presented results provide valuable insight into North Sea low-level jet characteristics. These jets occur predominantly for circulation types with an easterly component, with a clear peak in spring, and concentrate along the coasts at heights between 50-200 m. Further, it is demonstrated that these characteristics can be used as predictors to infer the observed low-level jet rate from ERA5 data with reasonable accuracy.

*Copyright statement.* TEXT

# 1 Introduction

On average, wind speed increases with height above the surface and the rate of increase can be described using simple formulas (e.g. power-law or logarithmic profile, see Sedefian, 1980). Due to their simplicity and ease of use, these *wind profile parameterizations* have been widely adopted in the wind energy community. However, in some situations these formulas cannot

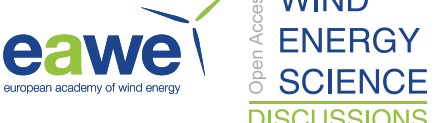



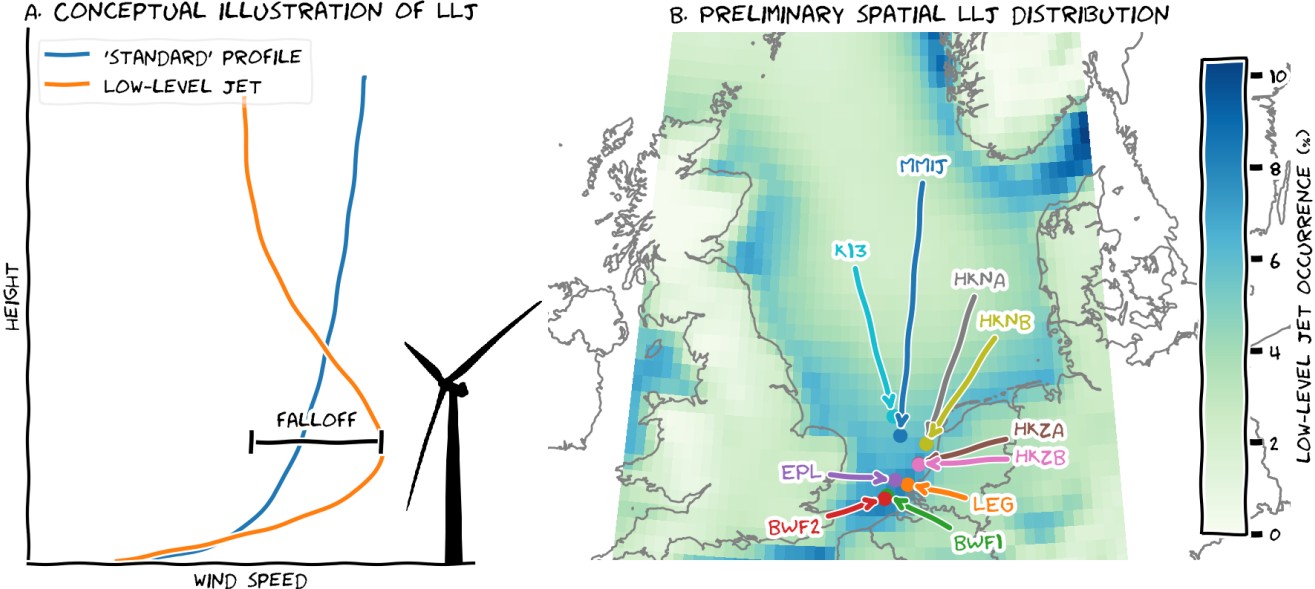

**Figure 1.** A. Example low-level jet profile as compared to the 'standard' logarithmic wind profile. B. Preliminary spatial distribution of annual low-level jet occurrence based on 10 years of ERA5 data up to 500 m. Overlaid are the location of the 10 measurement platforms used in this analysis: Met Mast IJmuiden (MMIJ), Hollandse Kust Noord A (HKNA) and B (HKNB), Hollandse Kust Zuid A (HKZA) and B (HKZB), Lichteiland Goeree (LEG), Borssele Wind Farm Lots 1 (BWF1) and 2 (BWF2), Europlatform (EPL) and K13. Color coding is consistent across all figures.

adequately capture the observed wind profile. During these situations, application of a simplified wind profile parameterization can introduce error or 'uncertainty' into the reconstructed wind climatology. This is clearly the case for low-level jets, for which wind speed reaches a maximum not far (i.e. roughly less than 500 m) from the surface (Figure 1A). Wind shear and turbulence intensity associated with low-level jets also differ substantially from that assumed under 'standard' conditions.

5     Low-level jets modify wind power performance and loading by impacting wake recovery rates and vertical profiles of wind speed, direction and turbulence (Wharton and Lundquist, 2012; Bhaganagar and Debnath, 2014; Park et al., 2014; Gutierrez et al., 2017). Thus, for a complete assessment of loads and power, it is important to have a broad understanding of the site-specific low-level jet characteristics: how often do they occur, under which circumstances, at what height and with what strength, and what mechanisms are responsible for their formation? While some studies report on low-level jets in coastal areas

10   (e.g. Nunalee and Basu, 2014, especially section 2.3 and references therein), a systematic long-term characterization is lacking for the North Sea.

    In a previous publication (Kalverla et al., 2017), we reported on low-level jet characteristics at a prospective wind power site 85 km off the Dutch coast (MMIJ, aka "IJmuiden ver"), using 4-years of mast and LiDAR observations. The climatology consisted of: the diurnal and seasonal variability in low-level jet occurrence, jet speed, jet height, jet direction, et cetera.



Inherently, this low-level jet climatology is only valid for the single observation site examined. In order to generalize the results from this study, and to improve our overall understanding of low-level jets across the North Sea, we now present a spatial climatology of low-level jets based on ERA5 (Section 2; Copernicus Climate Change Service (C3S), 2017) reanalysis data and an extended set of observations.

Preliminary results based on 10 years of data in the lower 500 m of the atmosphere (Figure 1B) shows that ERA5 provides interesting information about the spatial distribution of low-level jets. However, without observational support this information is of little value. Therefore, we incorporate additional LiDAR observations to provide this support, but knowledge gained of the Dutch offshore wind climate from these measurements is inhibited by the relatively short duration of measurement collection (i.e. typically ∼ 1 year) and the limited vertical measurement range (i.e. typically less than 300 m; see Appendix A

for details on measurement time and range). Consequently, the aim of this study appears twofold: (1) observations will be used to validate the ERA5 climatology of wind and low-level jets and (2) ERA5 will be leveraged to infer long-term low-level jet characteristics based on a limited set of observations. Absolute agreement in low-level jet characteristics between the two data sources would enable perfect execution of these objectives; however, that is unlikely. Therefore, we formulated the following research question serve/blend both perspectives:

*How can observations and reanalysis data be combined to obtain a spatial climatology of low-level jets that is both rich (in its spatial and temporal extent) and reliable (in terms of its correspondence with available in-situ observations)?*

The paper is structured as follows. A brief description of the data and an elementary evaluation of wind speed itself is provided to illustrate how both datasets are biased. Thereafter, low-level jet representation within both datasets is discussed, starting with jet detection and morphology (e.g. jet height). A common thread throughout the paper is how these characteristics

are impacted by time and (vertical measurement) range limitations. Using the seasonal cycle of low-level jets as an illustrative example, we experiment with various methods to post-process the ERA5 data and extend the observations based on identified correspondence and/or differences. This exercise is repeated for the diurnal cycle, atmospheric stability and various circulation patterns. Finally, all of these characteristics are combined to demonstrate that the 'true' low-level jet rate can be reconstructed with reasonable accuracy if sufficient observations are available. The paper ends with a comprehensive discussion of the

implications and future research directions.

The focus of this paper is to obtain a reliable spatial representation of the low-level jets. This provides clues as to the physical mechanisms that govern them, but a detailed treatment of these processes is outside the scope of the current work.

To facilitate transparency and reproducibility, a series of Jupyter notebooks is available as supplementary material to this paper. Consequently, some technical details are left out of the main text, that is intended as a pleasant and coherent treatise of

the major results.

## 2   A brief description of both datasets and their shortcomings

Observations are available from seven sites (Figure 1B). Three of these sites had two LiDARs operating simultaneously. The temporal span of measurements ranges from six months to over four years. Some of the LiDARs were placed in the vicinity





of existing wind farms, and are appropriately filtered to remove any potential wind farm wake effects. More information on quality control and post-processing of the LiDAR data can be found in Appendix A. The observations are available as 10-minute averages, but to facilitate comparison with ERA5, the data were converted to hourly averages.

ERA5 (Copernicus Climate Change Service (C3S), 2017) is the latest reanalysis dataset from the European Centre for

Medium-range Weather Forecasts (ECMWF). Re(trospective )analysis is the procedure of fitting a state-of-the-art weather model to historical measurements (e.g. satellites, weather stations, etc.) to obtain a long-term dataset that is both spatially and physically consistent and depicts the state of the atmosphere as it evolved through time. ERA5 is the successor of ERA-interim, and similarly ERA5 is expected to be widely used for wind resource assessment studies (Olauson, 2018). Compared to its predecessor, ERA5 has a finer horizontal grid of about 30 km and also enhanced vertical resolution. ERA5 is based

on a newer model version and moreover, provides output at hourly intervals, enabling a comprehensive analysis of sporadic features such as low-level jets. ERA5 data from the North Sea domain between 2008 and (end of) 2017 in the lowest 500 m demonstrates the ability of the model to resolve low-level jets (Figure 1B).

Before analyzing the morphology of these jets, we illustrate the limitations of both datasets concerning the representation of wind speed. Figure 2A shows averaged wind profiles for the grid points closest to each of the measurement locations. The

full lines represent all 10 years of ERA5 data[1], whereas the dashed lines indicate averaged wind profiles derived from data subsets, which only incorporate ERA5 data when observations are available. The full lines are all quite close together, while the data subsets exhibit a much larger spread. Variability between the full lines can be related to physical differences between sites (e.g. distance to coast). Dissimilarity between the ERA5 10-year datasets and the ERA5 data subsets indicates that, due to the limited time extent of the observations, the data subsets are not representative of the site climatology. At MMIJ, wherein

measurements occurred for the longest period, this *representativity bias* reaches upwards of 1 m s$^{-1}$. The primary reason for this bias is that the MMIJ data contains more winter than summer months, and the wind is generally stronger in winter. Because the MMIJ data span more than 4 years, data can be discarded in order to ensure an equal representation of the seasons within the data. However, at the other stations, the temporal period of observation is limited. Therefore, implementing the same data modification techniques would result in almost half of the data being removed, which is not desirable. Worse still, HKN

observations do not encompass a complete year, and even if they did, inter-annual variability can be substantial. Available observations therefore cannot be used to derive the long-term wind climatology directly. However, by correlating a short-term dataset with long-term observations at a nearby site, the long-term wind characteristics at the target site can be inferred with reasonable accuracy. This procedure is known as measure-correlate-predict  (MCP, Carta et al., 2013). While not discussed here, application of similar techniques to the low-level jet phenomena will be examined later in this document.

ERA5 also demonstrates bias in its representation of site winds. An *error diagram* of the wind speed in ERA5 versus observations is provided in Figure 2C. In this diagram (co-opted from Kalverla et al., 2018), the mean error (bias) is plotted on the x-axis, the standard deviation of the error is plotted on the y-axis, and the distance to the origin represents the root mean square error (RMSE). Wind speed data from all observation levels were aggregated in this figure to evaluate the overall performance of ERA5 at each measurement site. The sites with the largest bias (i.e. systematic error) have the smallest standard

---

[1]Some lines are exactly on top of each other because they are in the same grid point. Both are plotted, though, to preserve color coding


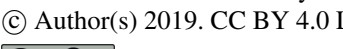


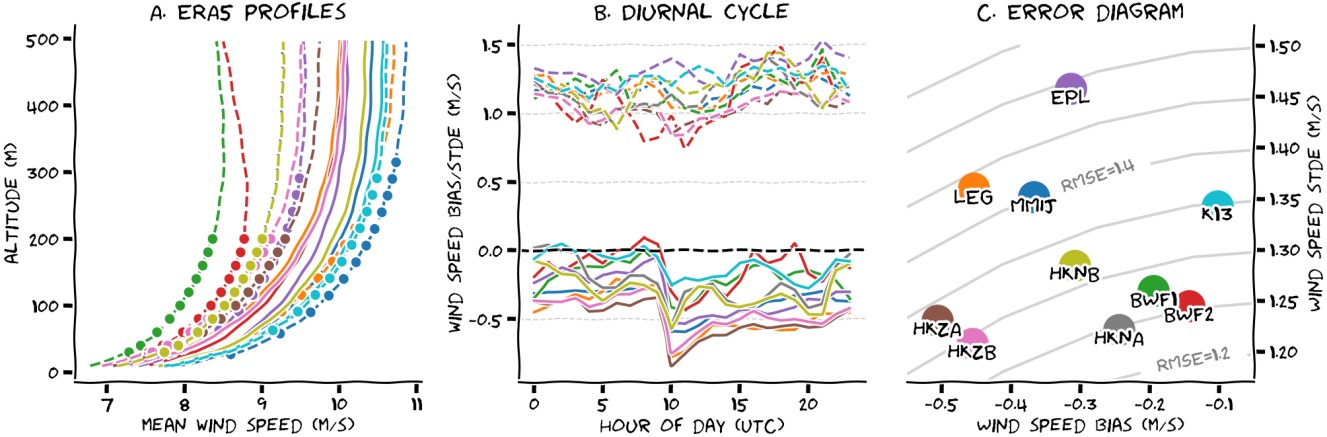

**Figure 2.** A. Averaged wind speed profiles for each measurement location, based on 10 years of ERA5 data (full lines) and data subsets (dashed lines). B. Mean (full lines) and standard deviation (dashed lines) of the error between ERA5 and the observations, for each measurement site, as function of the time of the day. C. Error diagram of wind speed in ERA5 versus observations for all LiDAR datasets. The color coding is the same throughout all subplots, so C can serve as legend.

deviation (i.e. random error). ERA5 site-specific RMSE values, ranging from 1.25 to 1.5 m s$^{-1}$, can be caused by multiple model aspects such as the limited grid resolution and the incomplete representation of physical processes.

The observed biases exhibit a strong diurnal variation. During the night (Figure 2B), the bias is roughly between 0 and -0.5 m s$^{-1}$, depending on the location. However, at 10 UTC, there is a sharp decrease in the bias, downwards of -0.8 m s$^{-1}$ for some

stations. The reason for this discontinuity can be found in the IFS (Integrated Forecasting System) documentation (ECMWF, 2016). ERA5 is produced with a 4D-VAR data-assimilation algorithm that uses two 12-hourly windows running between 9-21 and 21-9 UTC. This means that all hourly fields up to the 9 UTC analysis are based on the nighttime observations, while data from 10 UTC onwards are based on the daytime observations. We hypothesize that the impact of the data-assimilation is magnified during the nighttime, because nighttime boundary layers are generally shallower. Discontinuity in the diurnal cycle

is present at each model level up to 300 m, irrespective of the season and platform; however, it seems to be slightly stronger for those stations closer to the coast.

## 3 Jet detection: a precarious procedure

Low-level jets are identified by seeking local maxima in the wind profiles. Having identified a local maximum, the jet strength, height and falloff are analysed. Falloff, as indicated in Figure 1A, is defined as the difference between the maximum and the

subsequent (moving upwards) local minimum or, if no local minimum is present, the top of the wind profile. Most results in this study are based on an absolute falloff threshold of 2 m s$^{-1}$. Figure 3 demonstrates how this threshold influences the low-level jet detection rate, and further how the detection of low-level jets is influenced by both time and (vertical measurement) range



limitations. The figure consists of five scatter plots, each depicting the falloff versus the jet height for each wind profile that was detected with a local maximum. The differences between the panels are the underlying data analysed - i.e. observatings and varying subsets of ERA5 data.

The first panel (Figure 3A) is based on 10 years of ERA5 data and the model levels contained within the lower 500 m of the
atmosphere. The two dashed lines represent limiting factors: (1) the falloff threshold of 2 m s$^{-1}$ (horizontal dashed line) and (2) limitations due to observation height (vertical dashed line). The model data extend up to 500 m, but the observations reach only up to about 300 m (depending on the platform). All platforms are overlaid (shorter datasets on top). Only points above the horizontal dashed line are included in the low-level jet climatology that is presented in the next sections. Figure 3B-D are based on subsets of the ERA5 dataset. In panel B, ERA5 data is incorporated only if observations are available; as expected, this
substantially limits the total number of low-level jets. In panel C, we have retained all 10 years of data, but only at observation heights[2] (i.e. data above 300 m were discarded and the remaining data were vertically interpolated – using a cubic spline – between the remaining model levels to obtain the ERA5 wind speeds at the exact observation height). This effectively filters out all meaningful jet events from the ERA5 data, not just those above 300 m. In order to classify a wind profiles as a jet, falloff above must be properly resolved. This explains why a jet at 100 m can also vanish from the climatology if data from above
300 m are removed. The pronounced impact of this vertical range limitation on the ERA5 data raises the question whether the *observed* low-level jet climatology would be much different if we could observe higher-altitude winds. Increased measurement range might reveal not only low-level jets above hub-height, but also new low-level jets at hub-height that are currently not identified as such.

Height and time limitations are combined in panel D in order to develop an ERA5 dataset that is fair to compare with
observations (Panel E). Simple visual inspection indicates that ERA5 does not perform well. A contingency table based on one-to-one (1:1) jet correspondence between the two datasets shows a very low critical success index (∼0.2) and probability of detection (∼0.2). Does that imply that ERA5 is useless? No! Figure 3A indicates that potentially relevant information was filtered out. ERA5 jets might have been vertically displaced or potentially just not strong enough? This would not come as a surprise: weather models have long been known to generate excessive vertical mixing under stable conditions, effectively
'smearing out' low-level jets (Holtslag et al., 2013). If the height thresholds for the ERA5 data is modified to 500 m, the 1:1 correspondence is still quite poor (critical success index ∼0.2; probability of detection ∼0.5), but despite an inability to accurately denote total number of low-level jets, other characteristics appear to be captured quite well – e.g. the average monthly low-level jet rate. Therefore, the remainder of this paper is devoted to the analysis of such low-level jet characteristics and methods to consolidate ERA5 and measurement data.

---

[2]In contrast to the model level data, which display small variations in model level height, the observation level data are at fixed heights. To improve readability of the graph, we added jitter (small random deviations) to these heights.





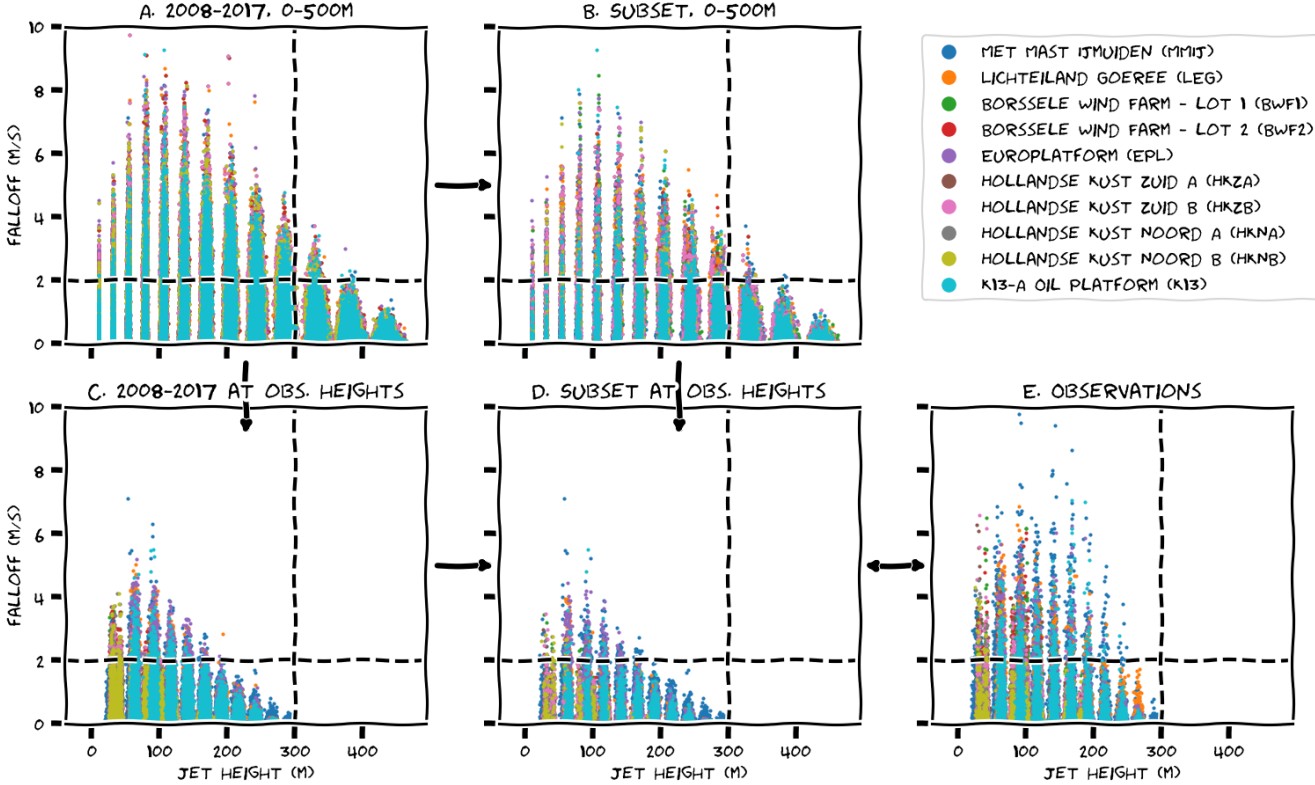

**Figure 3.** Scatter plots of falloff versus jet height for various representations of model data, and observations. See text for explanation.

## 4 Vertical range affects perceived jet morphology

Jet height and jet strength are of paramount importance for wind energy applications. Small variations in height can result in either symmetric or asymmetric loads on the turbine, and typical strengths in the rated part of the power curve are probably less critical than typical strengths in the cubic part. It turns out, though, that the concepts of 'typical' height and strength are

5 not self-evident.

Figure 4 displays probability distributions of jet strengths (panel A) and jet heights (panel B) for various representations of the ERA5 data and observations[3]. It shows that the jet height and strength distributions are sensitive to the range limitation. The median observed jet strength is about 8 m s$^{-1}$. This is quite well reflected in the ERA5 data if we consider all levels up to 500 m, but after imposing the range limitation, the jet strength is underestimated by about 3 m s$^{-1}$. The observed median jet

10 height is around 80 m. The ERA5 jet height distribution is broader with greater jet heights for the data up to 500 m, while it is narrower with lower jet heights for the range-limited data. To obtain a robust result, this figure is based on the aggregated data

---

[3]Obviously, it is physically impossible to have a jet strength or height below zero. This is an artifact of the visualization - it has a smoothing effect. We experimented with other visualizations (smaller bandwidth, or histograms), but found that this visualization best represented the underlying data.



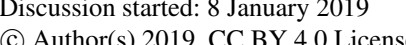

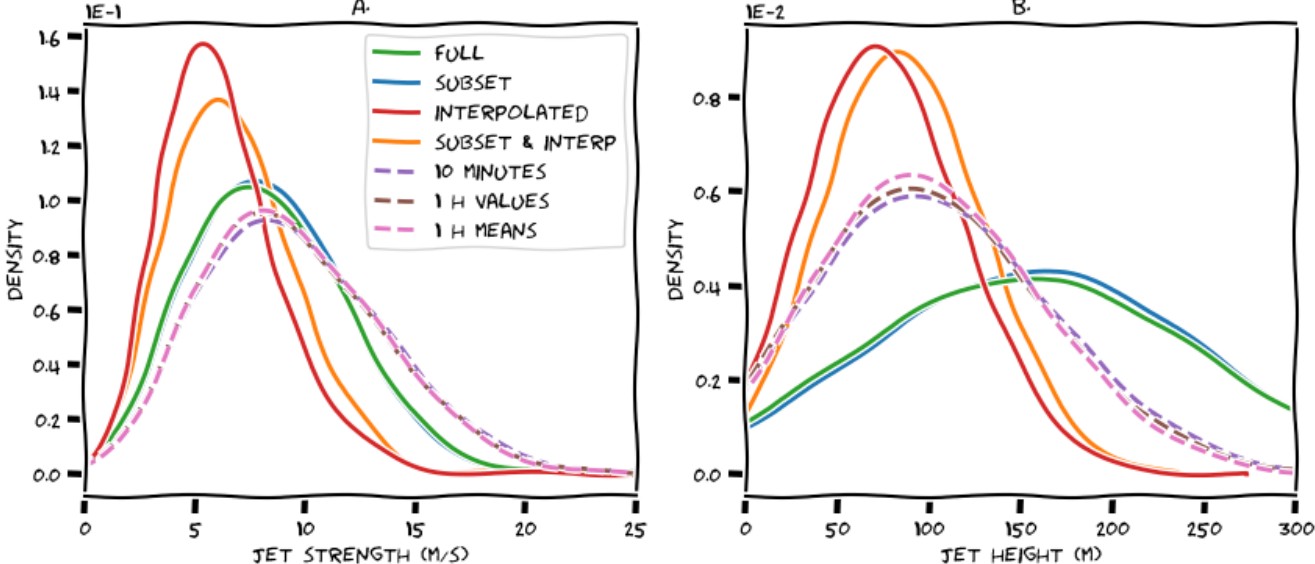

**Figure 4.** Kernel density estimates of the probability distribution of jet strength (A) and jet height (B) for various representations of the ERA5 data (full lines) and the observations (dashed lines), aggregated over all stations.

from all platforms. Separate figures for each individual platform show similar characteristics, although the jets near the coast seem to be somewhat closer to the surface than jets further offshore (not shown).

Three different representations of the observations are included in Figure 4. The first one is based on the 10-minute data. The second is based solely on the data of each full hour; in other words, we discarded 5/6th of the data. With this strategy, (small) discrepancies in low-level jet timing can have a disproportionate impact on the results. A more permissive evaluation is based on hourly averages obtained with a sliding window, where each full hour is an average including the 10-minute data from the preceding and the following two time stamps. This last version of the observations is used throughout the remainder of the paper. This figure demonstrates that the differences between various resampling methods in terms of jet height and jet strength are small.

## 5 Datasets agree: most jets in spring and summer

Figure 5 displays the seasonal cycle of low-level jets and, in a similar fashion as Figure 3, how this cycle is subject to time and range limitations. Over 10 years' time and 500 m (panel A), the seasonal cycle is smooth and differences between the individual platforms are small. Ideally, we would compare this to 10 years of observations up to 500 m, but since those data are not available we take spatial and temporal subsets of the ERA5 data instead. By investigating how this affects the seasonal cycle, we identify methods to extend upon the limited observations. Over the shorter measurement periods (panel B) the seasonal cycle appears much more erratic than the 10-year climatology. Some years are not very representative, and some





datasets do not even cover a complete cycle. As we will see later on, a favourable weather pattern for low-level jets is a weak large-scale forcing typically associated with high-pressure systems. Such 'blocked' weather patterns can last for several weeks, and their occurrence can thus cause large differences in monthly low-level jet rates. In other words, the seasonal cycle based on only one or a few years is very sensitive to inter-annual variability. Upon vertical subsetting/interpolation to measurement

heights (panel C) the seasonal cycle is still visible, albeit with a much smaller amplitude. The combined effect (D) leads to a very uninformative climatology, because the monthly low-level jet rates are all (close to) zero except for some unrepresentative spikes. Based on panel B, we expect that the observations are similarly affected by the limited time window of the observations. Indeed, panel E shows an erratic seasonal cycle with an amplitude somewhere between panels B and D.

Thus, both datasets agree on the presence of an annual cycle, but the amplitude differs between (various representations

of) ERA5 and the observations. Moreover, the observation periods are too short to obtain a reliable climatology. To distill a more robust signal from the observations, we combined all platform data, computed aggregated monthly means, and then used a moving average of 3 months to obtain a smooth signal (the dashed black line if panel E), which mostly differs from A in amplitude. We then repeated this exercise for all other panels and adjusted the amplitudes by scaling the resulting signal with the ratio between the mean of the ERA5 data and the mean of the observed cycle. The result is promising: the seasonal cycle is

similar for all datasets, peaking at about 5% in June. The crude manipulation of the data leads to a large error margin, though, and we wonder whether we can find a more sophisticated approach to achieve a similar result. Furthermore, because valuable information is lost if we discard the ERA5 data above observation heights, we will continue to work with the ERA5 data up to 500 m in the remainder of this paper.

## 6   Simple scalings for the seasonal cycle

In the previous section we learned that ten years of ERA5 data lead to a smooth seasonal cycle, but shorter observation periods lead to an erratic seasonal cycle because the months in the subset are not representative of the long-term monthly means. We also saw that upon aggregation and smoothing, both ERA5 and observations show similar seasonal cycles that differ mostly in their amplitudes. In this section we seek to combine the information from both data sources to reconstruct the 'true' seasonal cycle of low-level jets over the North Sea. We considered two different approaches.

The first method applies a correction to the observations, based on information about their representativity. For each month, we calculated the ratio between the low-level jet occurrence in the full- and subsets of the ERA5 data. If this factor is much smaller (or larger) than 1, the months in the subset are characterized by above(below)-average low-level jet occurrence. We then applied these ratios as correction factors to the observed monthly means to adjust the outliers and obtain a more representative seasonal cycle. However, this method did not lead to satisfactory results, because the correction factors were not robust: if

only 1 year of data was available, and a month was very unrepresentative, the correction factor would become very high/low and the adjustment would overcompensate. Consequently, the reconstructed long-term seasonal cycles still appeared erratic and were deemed unreliable (this result is therefore not shown here, but is available in SI 4/6). For MMIJ the measurement period spanned more than 4 years and consequently, the monthly low-level jet occurrence already started converging to the

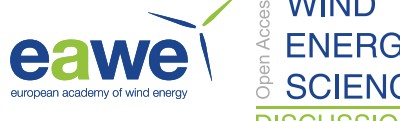

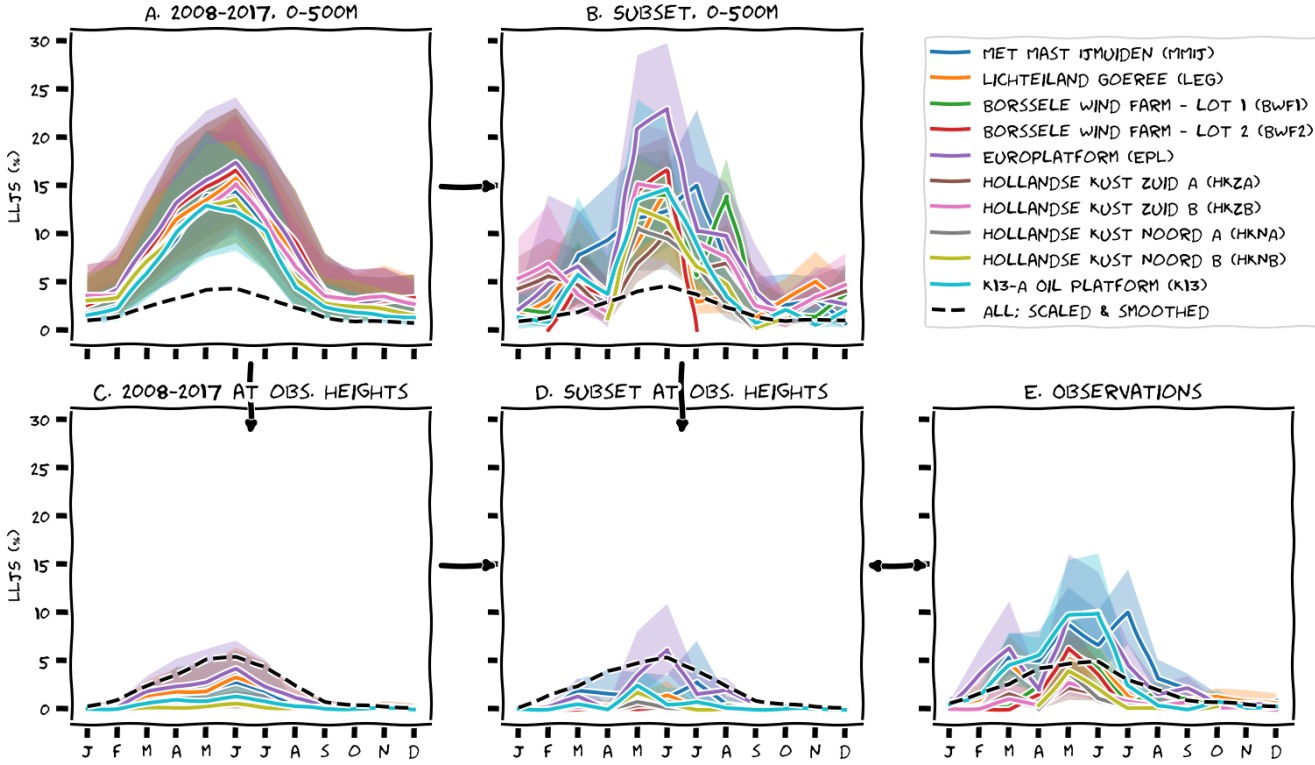

**Figure 5.** Seasonal cycle for various representations of model data, and observations. Shading is the sensitivity to +/- 0.5 m s$^{-1}$ for the LLJ falloff threshold. The dashed lines represents an aggregated seasonal cycle of all platforms, smoothed with a rolling average of 3 months (2 at the edges) and scaled with the ratio of the mean jet frequency in the respective representations of ERA5 and the mean jet frequency in the observations.

climatological seasonal cycle. For this platform, the correction factors were closer to 1 and we obtained a reasonably smooth seasonal cycle. This emphasizes that for this correction method, at least several years of measurement data are required obtain a reliable estimates of the long-term low-level jet climatology.

Whereas the first method was aimed at correcting the observations (using ERA5 as a 'vehicle' to assess their representativ- 5 ity), with the second method we aim to correct the long-term ERA5 data based on prior evaluation of its performance during the short-term period for which we have observations. This can be readily understood from Figure 5. We compare panels B and E, and seek a fixed scaling factor that minimizes their difference. Denoting the *monthly mean* low-level jet frequency in ERA5 and collocated observations with $\mathbf{x}$ and $\mathbf{y}$, respectively, an optimized scaling factor can be found by solving for $a$ in $\mathbf{y} = a\mathbf{x}$ (using linear least squares regression).

10    The results are illustrated in Figure 6A. The lighter colors represent fits to the monthly means for each individual platform, while the black line and scatter points represent the fit to the aggregated monthly means. This overall fit, based on all available data, has slope 0.44, but there are substantial differences between the individual platforms, with slopes between 0.15 and 0.73





and relatively large scatter. The difference between platforms could be random, due to the limited availability of measurement data, or systematic, in which case different sites need different scaling parameters. If the difference is random, the global optimum indicated by the black line in Figure 6A could do justice to all individual platforms, because it incorporates a much larger body of measurement data than any single-site regression. Applying this factor of 0.44 to the full ERA5 data (Figure 5A)

provides us with a smooth seasonal cycle with reduced amplitude. In other words, the seasonal cycle of low-level jets based on ERA5 data up to 500 m overestimates the observed cycle (based on measurement up to 300 m) by a factor of ∼2. However, as shown in Figure 6B, there seems to be a spatial dependence in the scaling factors with larger slopes away from the coast, implying that the different sites need different scaling parameters. In order to cross-validate the single-platform regressions, we need to split the measurement data in train and test datasets, but this poses a challenge. Like before, the data record at MMIJ

is long enough to obtain a reasonable prediction of the test data, but some of the other data records are very short and splitting them would leave e.g. only 3 months of training data, which obviously leads to very poor statistics, especially since there are hardly any low-level jets in winter. Without cross-validation more data is available for regression, but this introduces the risk of over-fitting and therefore quantitative evaluation will be biased. Qualitatively, the resulting seasonal cycles still appear erratic (Suppl. Inf. 4/6).

Thus, despite similarities between the datasets, it is not straightforward to either correct the observations using ERA5 representativity factors, or to correct the ERA5 data using a scaling factor derived from collocated observations. In this section, we used the seasonal cycle to obtained aggregated low-level jet characteristics (i.e. monthly means), but perhaps we can identify other characteristics that lead to better results.

## 7   Other jet characteristics and their scaling potential

### 7.1   Diurnal cycle and stability

After analyzing in-depth the seasonal cycle of low-level jets, we now briefly consider some other variables that describe relevant characteristics of the low-level jet climatology, starting with the diurnal cycle. Figures 7A-C are again similar to Figure 5, now only including the ERA5 data up to 500 m. It appears that the low-level jets occur throughout the day, but with a small dip around 11 UTC. Panels B and C, based on short temporal subsets, are so erratic that it is difficult to distinguish this diurnal

cycle by eye. After aggregating all platforms and smoothing the data (black dashed lines), we find that the observations and ERA5 agree on the general shape, but again, the magnitude differs. As in the previous section, we performed linear regression to identify scaling parameters. The difference is that the low-level jet frequencies on which the fits are based are now grouped by hour instead of by month. The scatter in this data is larger than for the seasonal cycle, but the spatial distribution of the fitting parameters is similar (not shown).

The second row in Figure 7 shows the relation between low-level jet occurrence and atmospheric stability (expressed by the bulk Richardson number based on the ERA5 surface data). Scatter points represent mean aggregated low-level jet frequencies over 50 stability bins. Both ERA5 and the observations agree that low-level jets are typically associated with stable stratification, although for some platforms in Panel D, there seems to be a substantial number of jets for unstable conditions as well. In





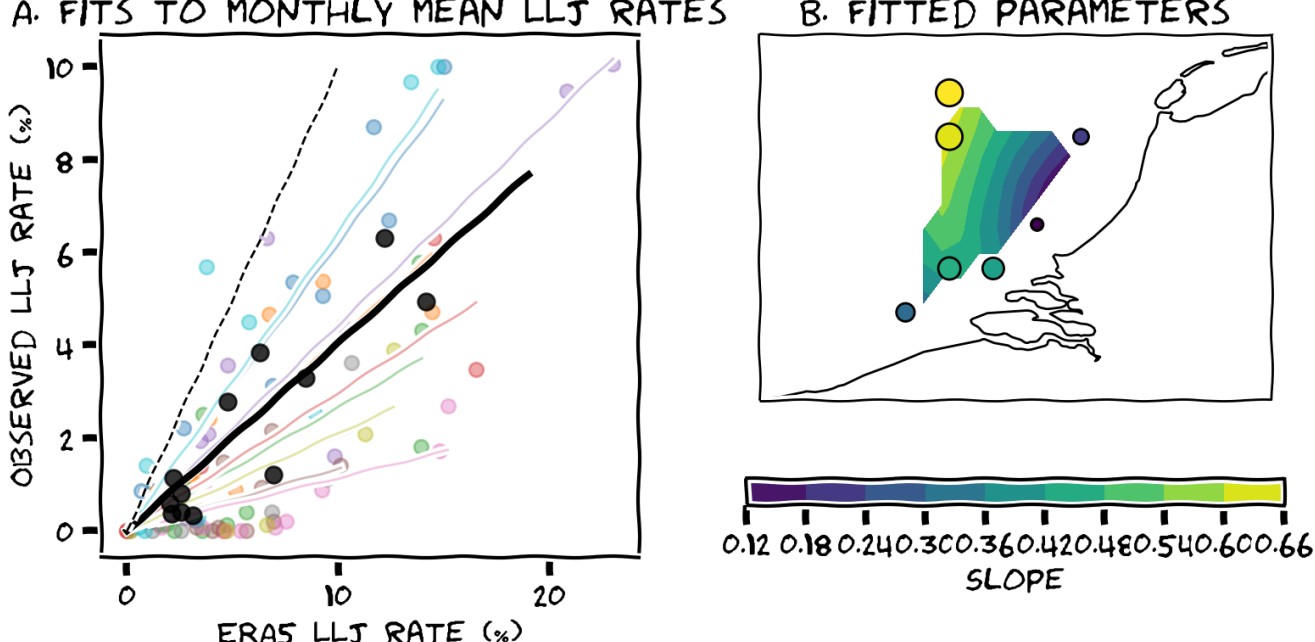

**Figure 6.** A. Illustration of linear regression between monthly low-level jet rates in the ERA5 data (subset, up to 500m) and the observations. Black line and scatter points represent aggregated data of all platforms, while the other colors correspond to fits for individual platforms. Dashed black line indicates a 1:1 correspondence. B. Spatial distribution of the obtained fit parameters for each individual platform. Like the color coding, marker size is scaled with the slope of the regression.

the subsets (panel E) this distinctive behaviour is not as clear, and in the observations it seems mostly absent. Without going into detail, we note that low-level jets can be formed by different mechanisms, and it is possible that ERA5 represents one mechanism better than another, or perhaps one mechanism is actually over-represented. Also note that in panels E and F there are (positive) values of the Richardson number for which no low-level jets are observed. In panel D, this is not the case, which

5  indicates that the measurement periods are too short to adequately sample the full range of stability conditions. Finally, we note that in panel D, the low-level jet rate seems to decrease again for very stable situations. This could be an artifact of the bulk Richardson number, or a physical limit: a stable atmosphere leads to a low-level jet, but the low-level jet produces wind shear and consequently, the bulk Richardson number decreases. The fact that this behaviour is not reflected in the observations suggests that the true stability (that would have been observed) was actually smaller than what ERA5 predicted. Again, we

10 tried to scale the amplitude of the stability signature by performing linear regression between the low-level jet frequency in ERA5 and observations (now aggregated over the stability bins). The slopes are larger than those based on the seasonal and diurnal cycle (∼1.0), but qualitatively they seem to be less robust.

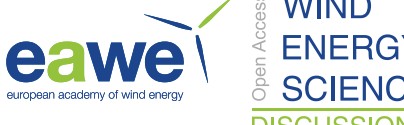

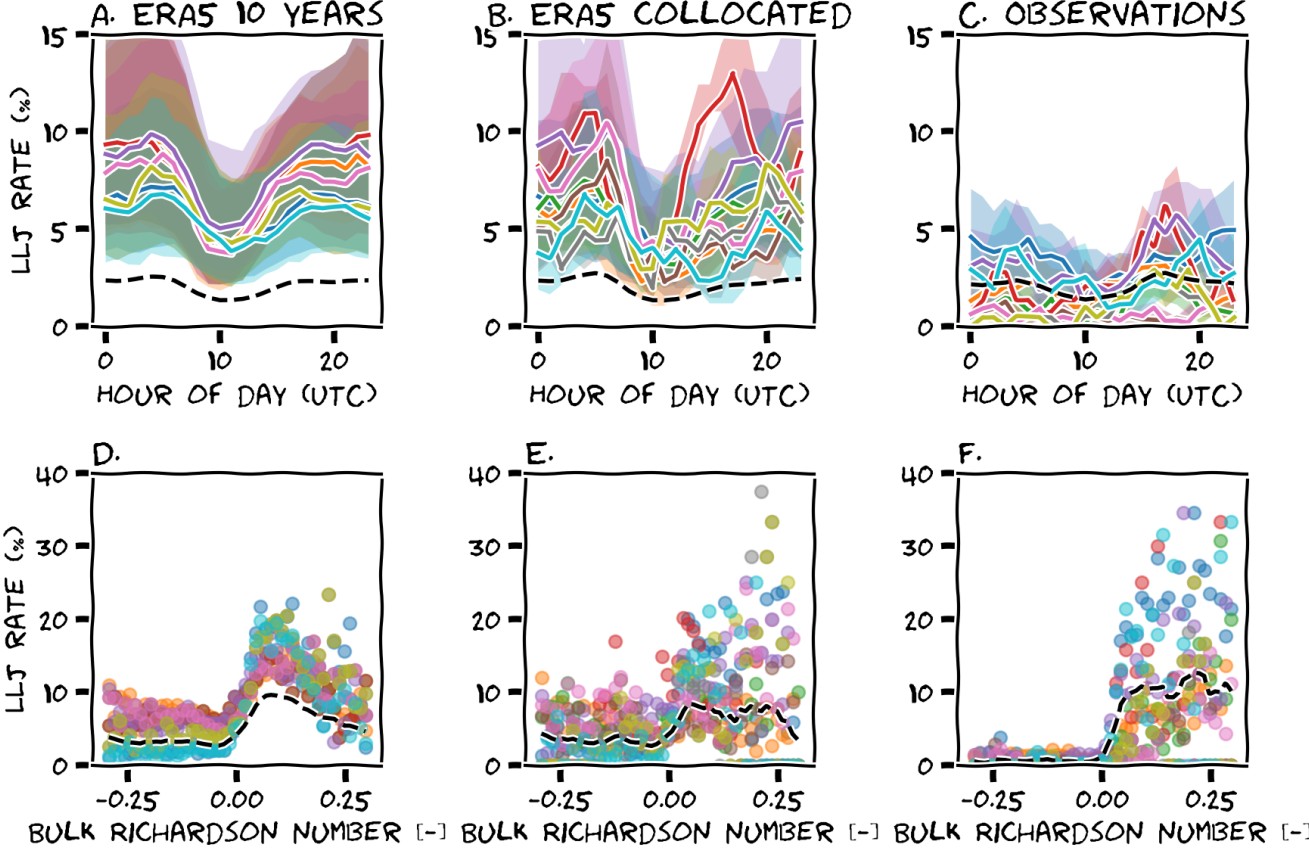

**Figure 7.** Average low-level jet rate for each hour of the day (A, B, C) and as function of the Bulk Richardson number (D, E, F), for the full (10 years) of ERA5 data up to 500m (A, D), a subset of this data collocated with the observations (B, E), and the observed data (C, F). Like in Figure 5, the black dashed lines represented a scaled and smoothed aggregated signal based on all platforms.

## 7.2 Weather types and the spatial distribution of low-level jets

We also investigated the relation between low-level jet frequency and typical circulation patterns. We used Lamb Weather Types (LWT; see Jones et al., 2013, especially the appendix) to perform this analysis. These weather types are based on gridded fields of mean sea level pressure data. The method distinguishes three main groups: those with a dominant cyclonic
5 (counterclockwise, low-pressure area) circulation, those with a dominant anticyclonic (clockwise, high-pressure area) circulation, and those with a 'pure directional' flow. These three groups are further subdivided based on the main direction of the flow over the North Sea (North, NorthEast, East, etc.). If there is no dominant direction, the LWT is 'pure (anti)cyclonic'. Pressure fields characterized by the absence of a dominant forcing are 'undefined'. In total this yields 27 different circulation patterns. We computed average low-level jet rates for each group.





To illustrate the association between the circulation type and the low-level jet occurrence, Figure 8 shows the average low-level jet rate per weather type in the North Sea domain, based on 10 years of ERA5 data up to 500 m. The streamlines show the dominant flow pattern for each weather type: the columns represent different wind directions over the North Sea, while the full rows represent different rotation types. In the first full row, the rotation is predominantly clockwise, in the bottom full row, the

rotation is mostly counterclockwise, and the middle full row is characterized by the absence of rotation. Notice how the same wind direction can be associated with different large-scale flows – and how this can impact the low-level jet rate. Like before, we will not go in-depth on each individual feature in this figure, but we will focus on overall characteristics. In general, we see that low-level jets concentrate along the coastlines, but they are much more dominant for certain Lamb Weather Types. Most notably, the 'undefined' weather type often gives rise to the formation of jets. This makes sense, as low-level jets are subtle

phenomena, and the absence of a strong large-scale flow eases their development. Furthermore, we observe that low-level jets occur frequently during large-scale flows with a pronounced easterly component. Note that easterly flows bring in continental air, while westerly flows originate from the Atlantic. Low-level jets are uncommon for westerly flows. Closer inspection reveals that the differences in spatial distribution of the low-level jets (e.g. comparing the Dutch, British and Norwegian coastlines) seems to be related to whether the large-scale flow is directed offshore.

Like with the previous characteristics, we performed linear regression between ERA5 and observed low-level jet frequency, this time aggregated over the various Lamb weather types. We found similar patterns in ERA5 and the observations (not shown), but the spatial distribution of the scaling parameters is different. Most slopes are around 0.4, but LEG stands out with a slope of 0.65. This is not a huge difference, but it implies that our earlier hypothesis – that the slope increases with distance to coast – does not hold for all predictors. Indeed, one could argue that with Lamb weather types as predictor, the scaling

parameters are spatially more robust. Thus, while we believe that the spatial distribution in Figure 8 is actually meaningful, the absolute low-level jet rates (as indicated by the color bar) is still off by a factor of $\sim$2.

## 8   Combining multiple predictors to extend observations

So far, we have tried to scale the low-level jet climatology with simple linear factors applied to individual characteristics (e.g. seasonal cycle). Perhaps, we can find a more sophisticated transformation function by combining multiple predictors?

In this section we use the MMIJ data to illustrate how this could be applied in practice. In contrast to the previous sections, which focused on aggregated low-level jet *frequencies*, here we consider *individual* wind profiles. The procedure resembles the Model Output Statistics (MOS) forecasts that are widely used for weather forecasts and is similar to the Measure-Correlate-Predict methods mentioned in Section 2. We use a machine learning package to perform this task and for readability, we will not highlight all the technical details here. However, Jupyter notebooks are available as supplementary material to facilitate

reproducibility.

The general idea is illustrated in Figure 9A: we have a short timeseries with observations and a long reanalysis dataset. Based on the overlapping part of the data, we determine the optimal parameters of a statistical model (depicted by the red box). We then use this model to predict the value of the observations, given the available long-term reanalysis data. In the illustration, it





**Figure 8.** Spatial distribution of low-level jet rate in ERA5 data. Amplitude is off by a factor of $\sim 2$ (best guess). A=Anticyclonic; C=Cyclonic U=Undefined; N, NE, E, etc are 8 wind direction sectors; combinations of a direction and a rotation type are 'hybrid' weather types, while weather types without a dominant rotational component are 'pure directional'. Streamlines illustrate the dominant large-scale flow pattern. The relative occurrence of each weather type is indicated as well. A larger version of this figure will be available online.



seems as though one reanalysis variable is used for this purpose, but in fact, we can use as many variables as we want. In our case, the variable we want to predict is the probability that a low-level jet will be observed, given various predictor variables from the ERA5 data. Because this is a binary outcome (a jet either occurs, or not), our model of choice is a logistic regression model, which predicts the probability of a positive outcome as function of one or several predictor variables. The general form of this model is

$$p = \frac{1}{1 + \mathrm{e}^{-(\beta_0 + \beta_1 x_1 + \beta_2 x_2 + ...)}}$$

where $\beta_i$ are the coefficients of the corresponding predictor variables $x_i$. In a short exploratory phase, we experimented with various combinations of predictor variables. We found the best performance for a small set of predictor variables consisting of: time of the year, atmospheric stability, and Lamb weather type. This makes sense, as together these variables encompass information about wind speed, direction and history of the flow, as well as the probability of stable stratification and baroclinic

conditions. Indeed, each of these variables alone already provided valuable information in the previous sections. For optimal performance, these variables were preprocessed as follows: to truthfully represent its cyclic nature, time was encoded by splitting the day of year in a sine and cosine contribution. The Lamb weather type is a categorical variable, and to make it suitable for regression it was encoded by converting it to the binary representation of the numbers up to 27 (the total number of weather types) and treating each digit as an individual binary variable. Stability was represented by the difference between the

two-meter temperature and sea-surface temperature, which provided better results than the bulk Richardson number. We also experimented with various training algorithms to determine the coefficients $\beta_i$ of the logistic model (intermediate results can be found in Supplementary Material). In the end, we settled on a stochastic gradient descend algorithm.

First, we took only half of the MMIJ dataset (a bit more than 2 years) to train the model (in other words: we fitted the parameters of our logistic regression model to the first half of the data). The light blue line in Figure 9B shows the seasonal

cycle of low-level jets in those first two years of observations. Notice that this seasonal cycle is very erratic; it must have had some very unrepresentative summer months. We then used our trained model to predict the other half of the dataset. The model predicts the probability that a low-level jet occurs. An individual jet is predicted only if the probability is higher than 50%, but this happens only occasionally. Therefore, rather than predicting individual low-level jet events, we reconstruct the predicted seasonal cycle by grouping and aggregating the predicted probabilities for each month (Figure 9B, orange line). To evaluate

the performance, we compared the predicted seasonal cycle with that based on the true observations during the second part of the dataset (Figure 9B, light green line). The true seasonal cycle was indeed smoother than in the first two years, but it peaked a bit higher and earlier than predicted. To quantify this result, we computed the root mean square error between the monthly means of the predicted and test-data, and found it to be about 1%. This result confirms that the model generalizes well to new input data.

We then used the full MMIJ dataset to train the same model. With twice as much training data as before, we are confident that the model will achieve at least similar performance and thus predict the seasonal cycle to within 1% RMSE (but probably better). The observed seasonal cycle averaged over these four years of training data (Figure 9B, red line) is still clearly affected by the unrepresentative months in the first half of the dataset. Apparently, four years of data is still not enough for the climatology to converge. Therefore, in the final step, we used the trained model to predict the 10-year seasonal cycle. The result





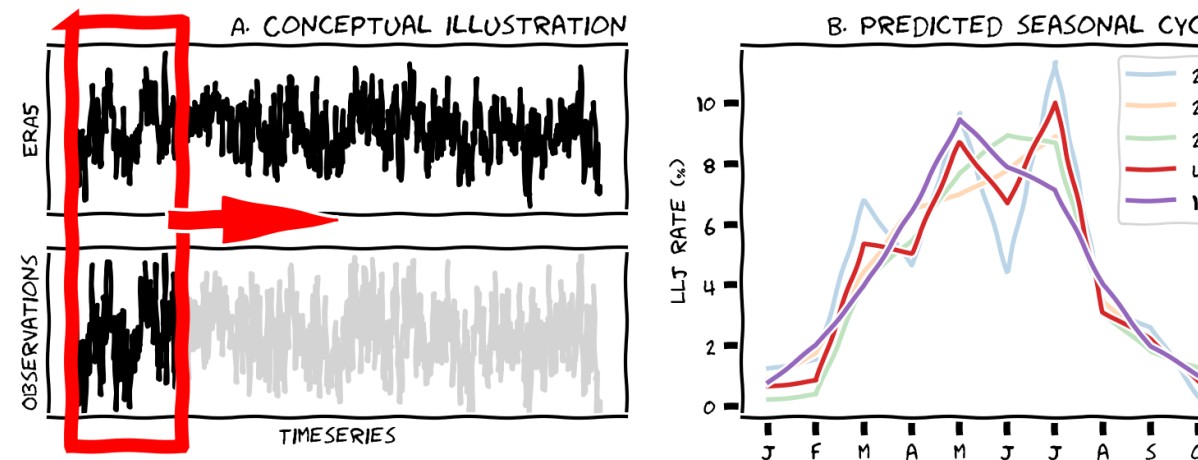

**Figure 9.** A. Illustration of the MCP/MOS/ML procedure in which a (logistic) model is trained with observation data and then used to predict long-term characteristics. B. Illustration of the MMIJ seasonal cycle of low-level jets based on: two years of observed data (train), two years of predicted (pred) and observed data (test), four years of observed data (train) and ten years of predicted data (pred).

(Figure 9B, purple line) is a smooth seasonal cycle which peaks in May at about 9%. This is our best estimate of the low-level jet seasonal cycle, based on the coalescence of reliable measurements and extensive reanalysis data.

The results presented in this section are intended as proof of principle, and for the purpose of illustration we tried to keep it conceptually simple. With respect to the selection of predictor variables, choice of model, and method of cross-validation, we realize that the possibilities are endless. The availability of sufficient measurement data is key to an exhaustive follow-up study.

## 9 Discussion

This paper has demonstrated our efforts to infer reliable low-level jet characteristics by combining observations and reanalysis data. We have deliberately chosen to illustrate how the results are impacted by limitations of the data and choices in the analysis. In this section we summarize our work, discuss the implications and offer an outlook to future research directions.

We started with a general validation of the ERA5 data for the observed wind speed at measurement locations at the North Sea. We found that the overall root mean square error is between 1.25 and 1.5 m s$^{-1}$. The bias shows a clear discontinuity at 10 UTC, which is related to the data assimilation strategy that was used to produce ERA5. Users of the ERA5 data should consider a suitable bias correction (e.g. Staffell and Pfenninger, 2016), but we strongly suggest that future reanalysis products use sliding or at least partly overlapping observation windows. We also demonstrated that the observations alone can neither be relied upon, because the limited temporal extent of the measurement data leads to biased climatologies. Thus, in the remainder of the paper we focused on finding a suitable way to combine the two datasets. A procedure similar to measure-correlate-predict methods but tailored to low-level jets instead.





The low-level jet detection is very sensitive to the vertical extent of the data, and this has important implications for the interpretation of all results. Typical jet characteristics like jet height and jet strength cannot be reliably inferred from range-limited observations. With this restriction in mind, we can say that many of the observed jets occurred at heights fully or partly in the range spanned by contemporary wind turbine blades. Moreover, typical observed jet strength is about 8 m s$^{-1}$,

which is in the cubic part of the power curves of these turbines. We therefore expect that the low-level jet impact on loads and power can be substantial. ERA5 is not able to reliably reproduce these characteristics. There are some indications that the jets are 'smeared out': they appear higher and weaker than observed. Given this vertical displacement, a fair comparison between ERA5 and the observations is difficult. Considering the lower 300 m only, ERA5 drastically underestimates the amount of jets, but including heights up to 500 m, ERA5 shows more low-level jets than observed. We decided to include the data up to 500

m because it gives a stronger climatological signature.

Even though 1:1 correspondence between ERA5 and the observations is poor, both datasets agree on the following climatological characteristics: most jets occur in spring and summer; the diurnal cycle is weak, only around noon the chances for low-level jets are slightly lower; low-level jets are typically associated with stably stratified conditions; the absence of a strong large-scale forcing, or flow regimes with a pronounced easterly or offshore component are favourable for their formation. From

the ERA5 data, we learned that low-level jets concentrate along the coasts. We then compared the frequency of low-level jets between ERA5 and the observations. In the most general terms, we can state that the mean low-level jet rates based on ERA5 up to 500 m typically overestimate the amount of low-level jets that would have been observed with LiDARs up to 300 m by a factor of about 2. To improve upon this result we then illustrated how a logistic regression model was able to predict the seasonal cycle of low-level jets at MMIJ to within 1% RMSE. This is a promising result, and we expect that our results can

still be improved upon. Longer measurement datasets would form a major contribution to further advancement as well.

The characteristics identified in this paper provide some clues as to the processes that govern these jets. The academic literature recognizes two dominant formation mechanisms, both of which are supported by our results. The first is frictional decoupling (Blackadar, 1957; Van de Wiel et al., 2010). This theory describes a perturbed system attempting to re-establish equilibrium. As the accelerating wind field in the lower atmosphere is deflected by the Coriolis effect, it moves about its new

equilibrium in a circular fashion. Over land, frictional decoupling has been linked to the decay of turbulent mixing around sunset and it has been suggested that a similar situation applies in coastal areas upon the abrupt surface (temperature and roughness) transition (Smedman et al., 1993). This mechanisms is supported by our results, which show that low-level jets are frequent for winds directed offshore and in stable conditions. The second mechanism relates low-level jets to horizontal temperature gradients (baroclinity, see Holton, 1967). According to this theory, the tilt of isobaric surfaces leads to a thermal

wind component that under certain conditions can manifest as a low-level jet. This mechanism has been coupled to low-level jets over gently sloping terrain, but equally applies to coastal areas where large horizontal temperature differences can occur due to differential heating between the land and sea surface (Mahrt et al., 2014). The fact that most low-level jets occur in spring and summer supports a baroclinic contribution, and possibly an interplay with the evolution of sea breezes, which show a similar seasonal cycle (e.g. Steele et al., 2015). In the end, we expect that both processes are likely to contribute to the

low-level jet climatology. Finally, we note that we also spotted a low-level jet with a clear frontal structure in the ERA5 data. It



is unlikely that such events contribute significantly to the low-level jet climatology, but the characteristics of such jets may be very different and potentially much more harmful for (offshore) wind turbines. Other causes have been described in literature, such as orographic blocking. We don't expect this plays a major role along the Dutch coast, but for some of the low-level jets that are present in ERA5 along the British and especially the Norwegian coast they may play an important role (Christakos et al., 2014). A more detailed investigation of the ERA5 data may allow us to separate these mechanisms. This is an interesting direction for further investigation.

With respect to future research, it would also be interesting to look at other datasets. In this paper, we have used ERA5 data to analyse the spatial characteristics of low-level jets directly. However, ERA5 is currently being used to develop higher resolution, down-scaled reanalysis datasets (e.g. the New European Wind Atlas (Petersen et al., 2013) and the Dutch Offshore Wind Atlas), and it would be worthwhile to see if they improve upon ERA5. Another interesting alternative is COSMO-REA6 (Bollmeyer et al., 2015), which is down-scaled from ERA-interim, but with its resolution of 6 km it might outperform ERA5. The current paper can serve as a guideline for the investigation of other reanalysis datasets.

Finally, a note on dealing with low-level jets in practice. It would be worthwhile to include a low-level jet case as standard inflow field for wake and load simulations. Recent papers have developed affordable methods to provide realistic inflow fields (Gebraad et al., 2014; Englberger and Dörnbrack, 2018). Expensive CFD simulations have been used to derive parameterizations to generate realistic inflow fields for wind farm simulations. The second cited paper also includes low-level jet profiles in the early morning. These profiles can be compared with the morphology and frequency distributions detailed in the current manuscript to optimize yield and lifetime. Since the presence of the coast line turns out to have an important effect on the formation of low-level jets, it would be interesting to perform an additional precursor LES simulation for such a heterogeneous terrain. This could also shed light on the mechanisms involved in jet formation.

*Code and data availability.* The ERA5 data were generated by ECMWF as part of the Copernicus Climate Change Service and will in the future be available through the Climate Data Store at https://cds.climate.copernicus.eu/#!/home. Observations were distributed by ECN part of TNO by order of the Dutch Ministry of Economic Affairs. They can be accessed at https://windopzee.net/en/home/. A series of Jupyter notebooks to facilitate reproducibility is available at github...

## Appendix A: LiDAR data

Vertically pointing LiDAR provides efficient and non-intrusive measurement of ABL winds. Compared to traditional meteorological masts, LiDAR typically expand the height and vertical sampling frequency of offshore wind measurements. LiDAR data from seven measurement sites were used in this study to analyse North Sea LLJ spatiotemporal behavior. LiDAR type used included the WINDCUBE v2 pulsed LiDAR (only at LEG) and the Zephir 300s continuous-wave (CW) LiDAR (all other platforms). The LiDAR were typically platform mounted, except within the Borssele wind farm and Hollandse Kust wind zones





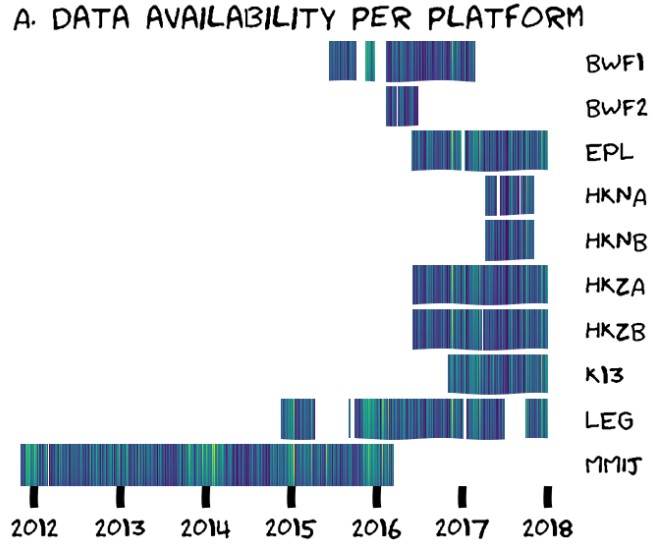
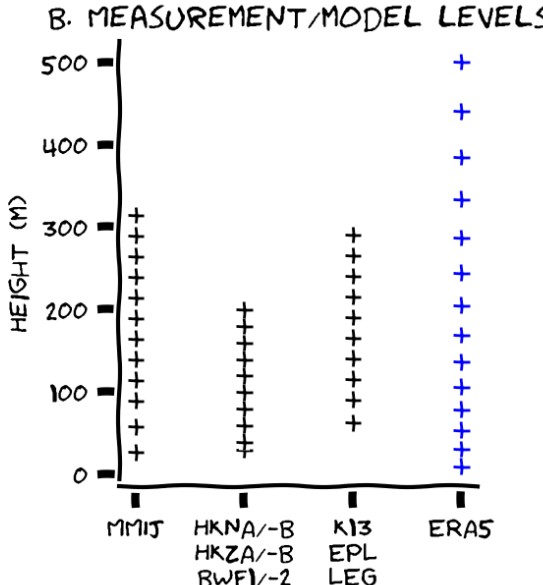

**Figure A1.** A. Time-height plots of wind speed for each platform, illustrating the data collection periods, temporal overlap between platforms and episodes of missing data. B. Site-specific measurement heights. Reference elevation for the ERA5 data have been included for comparison.

(Noord and Zuid) where the LiDAR was instrumented atop a floating metocean buoy. At these locations, two LiDAR-equipped metocean buoys were positioned simultaneously.

CW and pulsed wind LiDAR are coherent systems, meaning they both analyse Doppler shift frequencies to determine an estimate of the radial wind speed (Peña et al., 2015). However, radial velocity and vertical wind profile extraction techniques

differ between the two LiDAR types. Whereas pulsed wind LiDAR use range gates to near-simultaneously extract radial velocity estimates at multiple points in space, CW wind LiDAR can only extract a radial velocity estimate at the beam focus length. This beam focus length must be modified in time in order to measure the wind field at varying elevation levels. The radial wind speed is defined as the motion of the wind towards or away from the remote sensing system, and therefore unless the wind is moving along one of these radials, then the wind speed will not be fully resolved. Consequently, CW and pulsed

wind LiDAR use varying adaptations of conical scanning techniques (Banakh et al., 1995) to resolve the horizontal wind field at varying elevation levels. For brevity, these differences are not detailed here. However, because of these differences, the vertical wind profile was resolved at 17-s intervals for the CW wind LiDAR and at 4-s intervals for the pulsed wind LiDAR. These wind profiles are then analysed by the LiDAR software and outputted as a 10-min average vertical wind profile. A summary of the LiDAR measurement heights and data collection periods for all sites is provided in Figure A1.

Data quality control is imperative to ensure an accurate depiction of the offshore LLJ. Implementation of data quality control varied depending upon the LiDAR type (i.e. ZephIR 300s versus WINDCUBE v2), albeit considerations were made to ensure



that data quality control was employed relatively uniformly between measurement sites. Wind LiDAR data from both the Borssele wind farm and Hollandse Kust (Noord and Zuid) wind zones have additionally had quality control measures implemented by Fugro Oceanor. An overview of these quality control procedures can be found online (https://offshorewind.rvo.nl). The data quality control procedures implemented are as follows. First, plausible value checks were implemented on the wind data. Any 10-min observation that met the following criteria was removed from the data record:

1. The mean wind speed was either greater than the period maximum wind speed or less than the period minimum wind speed.

2. The mean wind speed was less than 0.05 m s$^{-1}$.

3. Turbulence intensity (TI) for the period fell below 0.10 % (i.e. 0.001).

4. At the measurement height, the value of TI was 10 standard deviations ($\sigma_{TI}$) greater than the mean ($\mu_{TI}$) TI value (i.e. $TI \geq \mu_{TI} + 10\sigma_{TI}$); $\mu_{TI}$ and $\sigma_{TI}$ were defined as the height-respective value for the entire data collection period. Because TI typically decreases with mean wind speed, this threshold was only imposed if the 10-min mean wind speed exceeded 4 m s$^{-1}$.

Specific quality control measures were also applied to the LiDAR wind data. Any 10-min observation that satisfied the following criteria were removed from the data record:

1. A LiDAR error code (e.g. 9998 or 9999) was reported.

2. The carrier-to-noise ratio (CNR) was less than -22 (the value of CNR provides a measure of signal strength [i.e. quality]). CNR was only outputted by the WINDCUBE v2 wind LiDAR.

3. Backscatter magnitude was less than 1e-5 or greater than 100 – backscatter served as a proxy for CNR for data reported by the ZephIR 300s LiDAR.

Prior analyses (e.g. Poveda and Wouters, 2015) demonstrate that the ZephIR 300s LiDAR can incorrectly measure wind direction by 180°. Analyses of wind data at MMIJ from 1 January 2012 through 1 January 2014 indicated that approximately 3.6% of the measured wind data exhibited this flow reversal. Although mitigation (i.e. removal) of this data is possible, it requires independent wind direction measurements from a collocated meteorological mast. Because mast data was not available at each site, these wind direction errors were not removed. However, ZephIR 300s lidar wind direction errors did not appear to impact the measured wind speed, which is the main focus of this paper. In order to account for the wake effect of neighbouring wind farms on wind speed measurements, wind direction sectors were filtered and corresponding data (wind speed and direction) were removed. A generous estimate of 20 km was used to denote the maximum wind farm wake length.

*Author contributions.* Data analysis and preparation of the manuscript have been performed by the first author, under the supervision of the third and fourth author. Processing of LiDAR data was performed by second author, who also wrote the text for the appendix.



*Competing interests.* The authors declare that no competing interests are present

*Acknowledgements.* The analysis was performed on the high performance computing facility offered by the Dutch Science Organization NWO (grant number SH-312-15). This work is part of the EUROS project (NWO/TTW research grant number STW-14158).




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
