# Peer review of "Low-level jets over the North Sea based on ERA5 and observations"

_Wind Energy Science, 2018_

## Referee Comment (RC1) · Anonymous Referee #1 · 31 Jan 2019

Review of 'Low-level jets over the North-Sea based on ERA5 and observations: together they do better', by Peter Kalverla et al.

General comments

To my knowledge the authors are the first to publish a spatial climatology of low-level jets over the North Sea. This feat alone is a welcome initiative given the importance of this area for the wind energy sector, which will only increase in the coming decades. Moreover, the authors combine the spatial and temporal coverage of state-of-the-art reanalysis data with an extensive set of lidar observations from ten offshore platforms. Carefully addressing the pros and cons of both the model data and the observations, as well as properly discussing the difficulties in issues like LLJ detection, they provide a highly relevant and balanced overview of LLJ characteristics in the area of interest.

The manuscript has a clear focus and the results are described to-the-point. Methods are valid and the interpretations justified. For readers interested in more details, or want to reproduce the results, the authors provide extensive supplementary material. Overall, in my view this work can be finalized after minor revisions.

Specific comments

p1,ln8: 'bias of 1m/s' Ambiguous. Clarify that this is compared to the long-term mean.

p4,ln10: Specify on what grid the ERA5 data was retrieved for the present study.

p4,ln2: Given the large variability in data availability, it would be worthwhile to have Fig A1 here in the main text.

p4,ln14: 'gridpoint closest to meas. location'. Is this the case for the entire manuscript? I guess spatial gradients in ERA5 cannot be a priori ignored off the Dutch coast, in particular when plotting subtle differences as in Fig 2C. From suppl. inf. I note that interpolation is used, but this is good to mention this explicitly in the main text.

p4,ln23: 'the same ... technique': unclear, please clarify

p4,ln30: 'The sites with ...' This is a too strong statement since is depends totally on the HKZ lidars: ignoring them would lead to an opposite statement. Scatter is large.

p5,ln1: Please add a (few) lines on the observation uncertainty. To what extent could this contribute to the scatter in Fig 2C? Is it all the fault of ERA5?

p5,ln5: Valuable observation!

p8,ln6: 'average ... time steps'. This adds up to only 50 minutes of data, not an hour. Typo? Also, in 1 line explain the third representation of Fig4 in the text

Sect 3: Nice section clearly illustrating the non-trivial character of LLJ detection!

Figure 3: For clarity reasons, I suggest not to plot every single LLJ event. Suggestions: 1) apply some form of colour coding depending on the number of events for a number

of falloff bins per height interval, 2) distinguishing between the various sites has little added value in this plot.

p9, l26: 'calculated the ratio ... ERA5 data'. Specify if this is done for each location (and each month) separately. Line 27: 'months': I don't understand the plural form in relation to 'this factor' in the same sentence. Do you mean 'Months for which this factor is much smaller/larger than 1 are characterized by etc.'? Please clarify.

p10, ln 7. 'fixed' clarify: the same for all stations, for all months, or both?

p14, ln13: Mention the low LLJ frequency off the British coast, even for offshore wind directions. Seems to behave differently that the continental coast.

p14, ln 13: Refer to the work of Ranjha et al. 2013, who demonstrate that this increased LLJ occurrence along coasts is a global phenomenon. Ranjha et al. 2013: Global distribution and seasonal variability of coastal low-level jets derived from ERA-Interim reanalysis, TELLUS A, https://www.tandfonline.com/doi/full/10.3402/tellusa.v65i0.20412

p16, ln23: '1%' ambiguous in case of frequency of occurrence. I guess, also given Fig 9B, it should be 1 percent point, meaning a relative difference of ∼10%. Please clarify

p18, ln 19: see comment p16, ln23

Technical comments

p4,ln5: typo near )

p5,ln4: 'downwards of' check formulation. I would say something like 'sharp decrease in the bias of ∼0.5 m s-1 for most stations.

p9,ln12: 'if' -> 'in'

p10, ln2: 'required ... estimates' correct the sentence

p10, ln 10: remove 'fits to the monthly means for'; 'each' -> 'the'

p11, ln17: obtained -> obtain

p11, ln25: 'smoothing' -> 'smoothing and scaling'

p11, ln26: 'magnitude differs'. This is a bit inconsistent in the text: the magnitude of the (scaled) dashed lines in Fig 7 corresponds quite well, but here you refer to the unscaled cycles. Please clarify.

p11, ln 31: specify over what depth the bulk Ri was determined. Lowest observational level and something equivalent for ERA5?

p11, ln 33: 'Panel D' -> 'Panel D/E'

p12, Fig 6b: prevent tick labels in legend from overlapping

p14, ln 21: correct 'rates is'

p14, section 7: Given the fact that LLJ occur predominantly in spring/summer, but also for certain Lamb weather types, it would be interesting to see if these particular weather types also have a preferential occurrence in spring/summer.

p16, ln 7: 'day of year' > 'time of year'?

p18, ln1: remove 'The' at start of sentence

p18, ln24: 'about'??

p19, ln4: 'they' > 'this mechanism' or 'it'

p19, ln18: 'coast line' > 'coastline'

---

## Referee Comment (RC2) · Anonymous Referee #2 · 18 Feb 2019

**General comments:**

The manuscript provides an analysis of low-level jet (LLJ) statistics over the North Sea and combines reanalysis model data with actual observations from tall masts. A strength of the paper is that the authors point out the limitations of the respective datasets and show methods to overcome these limitations – some of them less, some more useful. Furthermore, the style and language of the paper is outstanding. The authors manage to explain methods and present results in a simple and clear language. At some places further clarifications would be helpful as detailed in the specific comments below.

I would like to highlight that the authors provide extensive supplementary material in form of a few Jupyter notebooks. These provide detailed information about how the

results were retrieved and allow readers to reproduce the analysis which makes the paper even more valuable.

A comment needs to be made about the graphics style. The authors use a special python package that revamps figures in a sketchy style. My personal opinion is that I like this style and I think it makes the paper much more attractive and accessible. While it is clearly beneficial for figures that show concepts and where exact curves or numbers are not intended or would even be misleading, it is questionable regarding really scientific graphs that illustrate results (measured or simulated). I assume the authors intend to convey that the results are to be seen with some uncertainty, and in this respect I would support the consistent use of the sketchy style. Perhaps it would be helpful to include a note about this? Otherwise, I believe that there will be some readers which will strongly object to this style and put the seriousness of the paper into question.

In general, I recommend the paper for publication after minor revisions.

**Specific comments:**

Page 2, lines 5-11: The literature review on previous LLJ studies, in particular LLJ statistics, is a bit short and should be extended.

Page 3, line 32: "Observations are available from seven sites (Figure 1B)." –> Make clear that only LiDAR observations are used, not met mast data.

Page 4, line 1-2: "More information on the quality control and post-processing of the LiDAR data can be found in Appendix A." Add "data availability" to the sentence –> "More information on the data availability, quality control and post-processing . . ."

Page 4, line 20: "At MMIJ (. . .) this representativity bias reaches upwards of 1 ms-1" –> In Fig. 2A the lines for MMIJ are only 0.4 m/s apart, not 1 m/s, and the bias for many

of the other sites is much larger. Please correct or clarify what you mean.

Page 4, line 30: "An error diagram of the wind speed in ERA5 versus observations" –> Which ERA5 dataset is meant: the full 10-year dataset or the subsets? Please clarify also in the caption of Fig. 2.

Page 4, lines 31-34 and Fig. 2C (error diagram): By definition and as also obvious from the figure RMSE and STDE are the same, aren't they? Your description of the figure and the figure itself suggest that there is a difference.

Page 5, lines 8-9: "We hypothesize (...)" –> Is there any literature available that could support your hypothesis?

Page 6, lines 4-18 and Fig. 3: Can you add numbers to Fig. 3 (and/or to the text)? By how much is the data reduced from A to C or B to D? It seems as if even below 300 m much more than 50 % of the data is removed.

Page 6, line 20: "Simple visual inspection indicates that ERA5 does not perform well." –> Give more details, e.g. similar height distribution but much smaller falloffs.

Page 6, lines 20-22: "A contingency table (...) shows a very low critical success index (...) and probability of detection (...)" –> Explain what this means, maybe also show the table.

Page 6, line 27: "other characteristics appear to be captured quite well" –> Add "e.g. the distribution of LLJs with height". Footnote 2: "In contrast to model level data (...)" Elaborate on this: why do the model level heights vary (is clear to me but maybe not to every reader)? How does adding jitter work?

Figure 3: Explain why the points are organized in these "bands". I assume this is due to the discrete model levels which vary in height.

Page 9, line 14: "the ERA5 data" –> The full dataset (A) or the subset (B)?

Page 9, line 10-18: Please clarify this procedure a bit more. It is hard to follow.

Figure 5A: Is is unclear if the dashed line in Fig. 5A is derived by the procedure described on p.9, l.13-14 or by the procedure described on p.11, l.5.

Page 10, lines 6-9: Clarify that each pair of monthly observed and simulated LLJ is considered. And clarify that all sites are taken together so that you obtain one single scaling factor for the combined dataset.

Page 11, lines 23-24: "It appears that the low-level jets occur throughout the day, but with a small dip around 11 UTC." –> The dip is not so small, the LLJ probability is significantly reduced between 8 and 16 UTC. Do you have explanations for this diurnal cycle and what does the literature say?

Page 11, line 26 and Fig. 7A-C: "but again, the magnitude differs" –> In Fig. 7A-C the dashed lines have the same magnitude. Please clarify.

Page 13, lines 2-9: Please describe which area you used to determine the LWT – is it the area shown in Fig. 8? So on how many grid points is the LWT derivation based? I assume you are using the ERA5 sea-level pressure field? How do you obtain the streamlines: Are you averaging all situations belonging to one LWT?

Figure 8: "Amplitude is off by a factor of 2 (best guess)?" –> What does this mean? It becomes clear from the text, but I would recommend to omit this information in the figure caption.

Page 14, lines 23-30: Can you give references for this type of procedure?

Page 16, line 15: "Notice that this seasonal cycle is very erratic" –> That is not surprising as it is only based on two years of observations.

Page 16, lines 18-19: "we reconstruct the predicted seasonal cycle by grouping and aggregating the predicted probabilities for each month" –> Please give more details.

Page 17, lines 1-2: "This is our best estimate of the low-level jet seasonal cycle (. . .)" –> Please link to the results in section 6 (which give a similar results).

Section 9: Very nice summary of the paper!

Figure A1A: What does the colour coding mean?

**Technical corrections:**

Page 6, line 2: "observatings" –> "observations"

Page 6, lines 12-13: "This effectively filters out all meaningful jet events (. . .)" –> I guess you mean "This effectively filters out all **not** meaningful jet events (. . .)"?

Page 6, line 13: "wind profiles" –> "wind profile"

Page 9, line 12: "(the dashed black line if panel E)" –> replace "if" by "in"

Page 10, lines 2-3: "are required obtain a reliable estimates" –> "are required to obtain a reliable estimate"

Page 12, line 12: Add "(not shown)" at the end of the sentence.

Page 16, line 15: "Notice that" –> "Note that"

Page 19, line 24: Please add the actual link of the Jupyter notebooks or mention that they are included as supplement to the paper.

Page 20, line 13: "outputted" –> "output"

Page 24, line 2: Kalverla et al. (2019): Add the correct volume number (currently "0").

---

## Author Comment (AC1) · 5 Mar 2019

Dear reviewer,

First of all, we would like to thank you for the time and effort you have invested in reviewing our manuscript. We appreciated your kind words and constructive feedback. In your specific comments, you highlighted numerous points in the manuscript where the formulation was unclear or imprecise. We have carefully addressed your concerns and prepared a revised version of the paper in which most of your feedback is implemented. Below, we have copied your specific comments and inserted our detailed response to each of your suggestions, including the modifications we made to the manuscript. With that, we trust we have adequately addressed your concerns.

[Figure]

Kind regards,

Peter Kalverla, James Duncan, Gert-Jan Steeneveld, Bert Holtslag

............................

Specific comments

p1,ln8: 'bias of 1m/s' Ambiguous. Clarify that this is compared to the long-term mean.

Response: We agree and implemented the suggestion in abstract.

p4,ln10: Specify on what grid the ERA5 data was retrieved for the present study.

Response: The ERA5 data was retrieved on a 0.3 degree lat/lon grid. We added this information.

p4,ln2: Given the large variability in data availability, it would be worthwhile to have Fig A1 here in the main text.

Response: We agree with the reviewer and placed this figure in the main text.

p4,ln14: 'gridpoint closest to meas. location'. Is this the case for the entire manuscript? I guess spatial gradients in ERA5 cannot be a priori ignored off the Dutch coast, in particular when plotting subtle differences as in Fig 2C. From suppl. inf. I note that interpolation is used, but this is good to mention this explicitly in the main text.

Response: The text is correct here, we used the nearest grid point. Only vertical interpolation was applied. Our rationale was that since we're considering heterogeneous terrain (especially if the next gridpoint is a land point with very different surface characteristics), interpolation might have adverse effects, and could even 'contaminate' the physical consistency of the model output. However, the reviewer brings forward a valid argument that spatial gradients can be strong, and therefore we checked whether interpolation would improve the results. We attach a figure illustrating the results. It appears that interpolation slightly improves the result, but the overall (rms) difference between

the two methods is only 0.04 m/s, or 3%. In figure 2c (now 3c), this error margin is within the size of the marker. Therefore, we prefer to stick with the 'pure' model output, and we added a comment about this comparison exercise at the end of this sentence.

p4,ln23: 'the same ... technique': unclear, please clarify

Response: We have modified this sentence to "and a similar seasonality filter would result in ..."

p4,ln30: 'The sites with . . .' This is a too strong statement since is depends totally on the HKZ lidars: ignoring them would lead to an opposite statement. Scatter is large.

Response: This is a fair point, we may have been a bit too eager in the description of this figure. We modified the text so that it now reads: "For example, the HKZ lidars show a strong bias (i.e. systematic error), but have a relatively small standard deviation (i.e. random error).

p5,ln1: Please add a (few) lines on the observation uncertainty. To what extent could this contribute to the scatter in Fig 2C? Is it all the fault of ERA5?

Response: We have added the following: "Uncertainties in the observations can also contribute to overall error statistics. Based on the manufacturer information and previous validation (Poveda, 2015), the uncertainty in the observations can only account for about 2% of the errors. Finally, displacement in space or time, as well as discrepancies between point-based measurements and modelled control-volumes can contribute to errors, although we've done our best to minimize these effects, e.g. by using appropriate time-averaging of the observations (see SI-II)."

p5,ln5: Valuable observation!

Response: We agree that this needs to be communicated.

p8,ln6: 'average ... time steps'. This adds up to only 50 minutes of data, not an hour. Typo? Also, in 1 line explain the third representation of Fig4 in the text

Response: Good point. Actually we used 50 minutes, because we wanted to center the moving average with equal weight before and after the full hour, but indeed... the observation data are 10-minute averages over the past 10 minutes, so we need to include one more record on the 'right' side. We made this modification in both the text and the analysis. The results are not affected. With respect to the second part of the comment: this is actually the third representation, but we didn't say that explicitly. Therefore, we modified the text to "A more permissive evaluation (the third representation) is based on ...."

Sect 3: Nice section clearly illustrating the non-trivial character of LLJ detection!

Response: thanks.

Figure 3: For clarity reasons, I suggest not to plot every single LLJ event. Suggestions: apply some form of colour coding depending on the number of events for a number of falloff bins per height interval, 2) distinguishing between the various sites has little added value in this plot.

Response: We acknowledge the concern about clarity and appreciate the suggestion to plot the data in a different format. While a 'hexbin' or 'density' kind of visualization provides a more quantitative view on the height-falloff distribution, it also hides certain features (e.g. it is no longer obvious that the underlying data is aggregated over multiple sites). Moreover, since the 'point density' in this figure has a broad range, the sparser (perhaps most interesting) areas become almost invisible (unless a non-linear colormap is used, which is perhaps even more confusing). Finally, from a practical point of view, we can no longer 'jitter' the data, which will result in a much more 'banded' and less clear figure. Considering that our main goal is to illustrate the jet detection procedure and not so much the exact number of individual low-level jets that are present in the data, we prefer to stick with the original formatting of this figure. However, to address the concern about clarity and to facilitate quantitative interpretation, we added the number of jets in each panel as well as the number of jets above the falloff threshold

in the top left corner of each panel.

p9, l26: 'calculated the ratio . . . ERA5 data'. Specify if this is done for each location (and each month) separately. Line 27: 'months': I don't understand the plural form in relation to 'this factor' in the same sentence. Do you mean 'Months for which this factor is much smaller/larger than 1 are characterized by etc.'? Please clarify.

Response: We modified "for each month" to "for each month and each location". And yes, this is exactly what we mean, and we agree that this formulation is much clearer, so we adopted it.

p10, ln 7. 'fixed' clarify: the same for all stations, for all months, or both?

Response: Two modifications. The first is "fixed scaling factor that minimizes their difference" is modified to "that minimizes the difference between each pair of monthly observed and simulated low-level jet frequencies." The second is that we added "We do this for each platform individually and also for their combined signal."

p14, ln13: Mention the low LLJ frequency off the British coast, even for offshore wind directions. Seems to behave differently that the continental coast.

Response: This is partly true, but if we consider figure 1b, the overall jet frequency here is not exactly low. It seems it is just less dominated by a certain weather type. Thus, we added that "The British isles are different in this respect, since for westerly flows, we do not observe an increased low-level jet rate off the eastern coast of the UK."

p14, ln 13: Refer to the work of Ranjha et al. 2013, who demonstrate that this increased LLJ occurrence along coasts is a global phenomenon. Ranjha et al. 2013: Global distribution and seasonal variability of coastal low-level jets derived from ERA-Interim reanalysis, TELLUS A, https://www.tandfonline.com/doi/full/10.3402/tellusa.v65i0.20412

Response: In the revised manuscript we have extended the literature review in Section 1, which now includes the study of Ranjha et al. At this point in the manuscript, we

added "In general, we see that low-level jets concentrate along the coastlines. This extends and refines the global findings of Ranjha et al. (2013) and Lima et al. (2018) for the North Sea domain.

p16, ln23: '1%' ambiguous in case of frequency of occurrence. I guess, also given Fig 9B, it should be 1 percent point, meaning a relative difference of âĹij10%. Please clarify.

Response: The reviewer is absolutely right, it should be percent point and we corrected this.

p18, ln 19: see comment p16, ln23

Response: corrected

―――――――――――――――

[Figure]

TIME/HEIGHT OVERALL DIFFERENCE WITH OBS

**Fig. 1.**

---

## Author Comment (AC2) · 5 Mar 2019

Dear reviewer

First of all, we would like to thank you for the time and effort you invested in reviewing our manuscript. We especially appreciated your general comment about the figure style. In a very considerate manner, you expressed almost exactly our own thoughts about the use of this style, including the hesitation. As suggested, we added a note at the end of the introduction stating that the consistent use of this style is in line with one of the main messages of the paper, i.e. to convey a notion of uncertainty. Your specific comments helped to improve the manuscript further and we have prepared a revised version in which most of your feedback has been implemented. To illustrate what we

have done with each of your suggestions, we have copied your specific comments below and inserted our response to each comment, explaining the modifications that we made to the paper. With that, we trust we have adequately addressed your concerns.

Kind regards,

Peter Kalverla, James Duncan, Gert-Jan Steeneveld, Bert Holtslag.

.........................

Specific comments

Page 2, lines 5-11: The literature review on previous LLJ studies, in particular LLJ statistics, is a bit short and should be extended.

Response: We already considered this for the initial submission, but at that time we decided to condense this paragraph in order to "get to the point". However, the fact that a reviewer now raises this point makes us come back on our initial decision, and therefore we extended the literature review again.

Page 3, line 32: "Observations are available from seven sites (Figure 1B)." –> Make clear that only LiDAR observations are used, not met mast data.

Response: this is not completely correct as in fact, met mast observations are included, but only at MMIJ. We added this information immediately after the first sentence of section 2.

Page 4, line 1-2: "More information on the quality control and post-processing of the LiDAR data can be found in Appendix A." Add "data availability" to the sentence –> "More information on the data availability, quality control and post-processing . . ."

Response: Another reviewer suggested to include the appendix figure in the main text. So instead of referring to the appendix, we now refer to this figure. With that, we think we have also addressed the underlying concern of this reviewer, namely that some information about data availability is appropriate at this point.

Page 4, line 20: "At MMIJ (. . .) this representativity bias reaches upwards of 1 ms-1" –> In Fig. 2A the lines for MMIJ are only 0.4 m/s apart, not 1 m/s, and the bias for many of the other sites is much larger. Please correct or clarify what you mean.

Response: Indeed, this must have been mixed up. We have modified the text to express that "for some stations, this bias reaches up to almost 2 m/s, and for MMIJ, for which the longest record is available, it still reaches up to about 0.5 m/s."

Page 4, line 30: "An error diagram of the wind speed in ERA5 versus observations" –> Which ERA5 dataset is meant: The full 10-year dataset or the subsets? Please clarify also in the caption of Fig. 2.

Response: this can only refer to the subsets, as we can only compute error statistics when observations are available. We have added the specification "(subsets)" in both this sentence and the figure caption.

Page 4, lines 31-34 and Fig. 2C (error diagram): By definition and as also obvious from the figure RMSE and STDE are the same, aren't they? Your description of the figure and the figure itself suggest that there is a difference.

Response: No, they are not the same, but they are related through $(RMSE)^2 = (STDE)^2 + (BIAS)^2$. The standard deviation of the error distribution is sometimes referred to as a 'centered' RMSE. We have added this relation to the text, since it is apparently not immediately obvious.

Page 5, lines 8-9: "We hypothesize (. . .)" –> Is there any literature available that could support your hypothesis?

Response: We agree that this would strengthen our argument, but unfortunately such papers are hard to find. While they don't support our hypothesis explicitly, we included the following references: "the difficulty of appropriately assimilating observational data within the (stable) boundary layer is discussed in Reen (2010) and Tran (2018)".

Page 6, lines 4-18 and Fig. 3: Can you add numbers to Fig. 3 (and/or to the text)? By

how much is the data reduced from A to C or B to D? It seems as if even below 300 m much more than 50 % of the data is removed.

Response: That's a good suggestion! We added the total number of jets as well as the number of jets exceeding the falloff threshold in the top left corner of each panel. This allowed us to make some quantitative statements in the text (e.g. "in going from panel A to C, 93% of the jets above falloff threshold vanished")

Page 6, line 20: "Simple visual inspection indicates that ERA5 does not perform well." –> Give more details, e.g. similar height distribution but much smaller falloffs.

Response: We agree that this statement was a bit vague. We based it mostly on the (underestimation of the) amount of jets (the height distribution is shown later). To clarify this, the sentence was changed to: "Judging from the figure, it seems that ERA5 does not perform well. Much fewer jets are found above the falloff threshold in the ERA5 data as compared to the observations. Indeed, a more quantitative comparison in the form of a contingency table ..." and then, after "was filtered out" we added a note that "even though the falloff is typically much smaller (to the extent that it falls below the falloff threshold), the height distribution of the ERA5 jets seems similar to the observations (also see Section 4)".

Page 6, lines 20-22: "A contingency table (. . .) shows a very low critical success index (. . .) and probability of detection (. . .)" –> Explain what this means, maybe also show the table.

Response: We added that "In other words, only 20% of low-level jets are correctly represented by ERA5". We considered showing the table here, but the number of choices involved in creating this table that would then also have to be explained and justified in the text would distract too much from the main focus of the paper. Note that this table is included in the supplementary information. Thus, we added a cross-reference instead.

Page 6, line 27: "other characteristics appear to be captured quite well" –> Add "e.g. the distribution of LLJs with height".

Response: See previous comment Page 6, line 20, where we already added statement about similar height distribution. With that, we think the current suggestion has also been addressed.

Footnote 2: "In contrast to model level data (. . .)" Elaborate on this: why do the model level heights vary (is clear to me but maybe not to every reader)? How does adding jitter work?

Response: Quite challenging to explain this in a footnote.. Here's our best try: "The ERA5 model levels are specified in terms of pressure rather than height, and can therefore exhibit small height variations in time. The observations, in contrast, are at fixed height, and to improve ...." However, see next response.

Figure 3: Explain why the points are organized in these "bands". I assume this is due to the discrete model levels which vary in height.

Response: Indeed. So we combined this comment with the previous suggestion and move the extended footnote to the figure caption. Hope this makes things clearer.

Page 9, line 14: "the ERA5 data" –> The full dataset (A) or the subset (B)?

Response: See next comment/response

Page 9, line 10-18: Please clarify this procedure a bit more. It is hard to follow.

Response: Indeed, we struggled a bit to formulate all these steps succinctly. To clarify, we have rewritten a few lines here. The new text is: "To distill a more robust signal from the observations, we combined the data from all sites before computing the monthly means, and smoothed the resulting signal with a moving average of three months. The result is the dashed black line in panel E. We then repeated these steps for the ERA5 data (panels A-D), but before plotting these lines, we scaled them with the observations, using a fixed scaling factor that is simply the ratio between the mean low-level jet rate in the respective representation of ERA5 (panel A-D) and the mean of the observations (panel E)."

Figure 5A: Is is unclear if the dashed line in Fig. 5A is derived by the procedure described on p.9, l.13-14 or by the procedure described on p.11, l.5.

Response: The figure is based on the first procedure, and the confusion is probably caused by our reference to figure 5a on p.11, l.5. We have modified the latter, and it now reads "Applying this factor of 0.44 to the full ERA5 data provides us with a smooth seasonal cycle with reduced amplitude (similar to the black dashed line in Figure 5A, but this time based on an optimized scaling factor)."

Page 10, lines 6-9: Clarify that each pair of monthly observed and simulated LLJ is considered. And clarify that all sites are taken together so that you obtain one single scaling factor for the combined dataset.

Response: Two modifications. The first is "fixed scaling factor that minimizes their difference" is modified to "that minimizes the difference between each pair of monthly observed and simulated low-level jet frequencies." The second is that we added "We do this for each platform individually and also for their combined signal." We adopted this terminology "pairs of monthly ..." also for the other sections: pairs of hourly, etc.

Page 11, lines 23-24: "It appears that the low-level jets occur throughout the day, but with a small dip around 11 UTC." –> The dip is not so small, the LLJ probability is significantly reduced between 8 and 16 UTC. Do you have explanations for this diurnal cycle and what does the literature say?

Response: Well, the dip seems big for the ERA5 data, but it is much less pronounced in the observations. Especially if you compare it with a land point (e.g. Cabauw), the diurnal cycle for the offshore platform is much less pronounced. In the meantime, we have further analyzed the ERA5 data, and it seems that there are at least two

mechanisms leading to the low-level jets: one related to the nocturnal jets onshore, and another leading to afternoon jets. This is probably related to the diurnal heating cycle, and especially the difference between land and sea. A third mechanism could be an inertial oscillation triggered by the coastal transition in offshore flows, but this would not have a diurnal signature. There is much more to say about these mechanisms and the literature, but we deliberately avoided going into too much detail here. To address the comment, we added that "From the observations, it appears that ..." and "The diurnal cycle in ERA5 is much more pronounced." And "At this point, we think it is good to stress that several mechanisms can lead to low-level jets in coastal areas (see Section1 and Section 9), and the diurnal signature should not be confused with that of the typical onshore nocturnal jet that is often found over land."

Page 11, line 26 and Fig. 7A-C: "but again, the magnitude differs" –> In Fig. 7A-C the dashed lines have the same magnitude. Please clarify.

Response: Modified to "but again, we needed to scale the ERA5 signals because they differed in magnitude."

Page 13, lines 2-9: Please describe which area you used to determine the LWT – is it the area shown in Fig. 8? So on how many grid points is the LWT derivation based? I assume you are using the ERA5 sea-level pressure field? How do you obtain the streamlines: Are you averaging all situations belonging to one LWT?

Response: We added "To derive these weather types we used the ERA5 mean sea level pressure on a 5-degree grid of 16 points as laid out in the appendix of Jones et al., but centered over the area of interest." And indeed, the streamlines represent averages, we added this to the figure caption.

Figure 8: "Amplitude is off by a factor of 2 (best guess)?" –> What does this mean? It becomes clear from the text, but I would recommend to omit this information in the figure caption.

Response: We understand that this information is confusing; on the other hand, we think it is necessary to warn readers who only scroll through the figures that this amplitude should not be taken for granted. Therefore, instead of omitting this information, we modified the warning to "As explained in the text, the values shown here overestimate available observations and should be interpreted with caution."

Page 14, lines 23-30: Can you give references for this type of procedure? Response: We included the following references. Carta et al. (referenced in section 2) gives a nice overview of MCP. MOS forecasts are commonplace in meteorological textbooks, so we referred to Wilks, 2006 (chapter 6.5.2), and additionally to two early papers (Glahn, 1972 and Carter, 1989).

Page 16, line 15: "Notice that this seasonal cycle is very erratic" –> That is not surprising as it is only based on two years of observations.

Response: True, but we still want to point it out. To clarify this, we added "Note that this seasonal cycle is very erratic. This can be expected for such a short period, but the question is whether the additional information contained in the predictor variables enables us to predict the other two years despite the unrepresentative training data. Thus, in the next step, we used our trained model ..."

Page 16, lines 18-19: "we reconstruct the predicted seasonal cycle by grouping and aggregating the predicted probabilities for each month" –> Please give more details.

Response: We agree that this is not very clear. Modified to "rather than predicting individual jet events, we used the predicted probabilities directly and computed the monthly mean predicted probability."

Page 17, lines 1-2: "This is our best estimate of the low-level jet seasonal cycle (. . .)" –> Please link to the results in section 6 (which give a similar results)

Response: We added "Compared to the results presented in section 6, we can conclude that we have adjusted the erratic nature of the short-term observations (Figure

5E), resulting in a seasonal cycle similar to that shown in Figure 5A, but with reduced amplitude. Compared to this final result, the crude amplitude adjustment with which we started in Section 6 now appears far too strong."

Section 9: Very nice summary of the paper!

Response: thanks. We hope that the discussion of mechanisms here also helps to address the reviewers concern about the (literature on) characteristics of the diurnal cycle.

Figure A1A: What does the colour coding mean?

Response: Added a short note "the color coding highlights episodes of high (yellow/green) and low (blue) wind speed."